# Transplantation of Fibroblast Sheets with Blood Mononuclear Cell Culture Exerts Cardioprotective Effects by Enhancing Anti-Inflammation and Vasculogenic Potential in Rat Experimental Autoimmune Myocarditis Model

**DOI:** 10.3390/biology11010106

**Published:** 2022-01-10

**Authors:** Kaori Sekine, Akira T. Kawaguchi, Masaki Miyazawa, Haruo Hanawa, Shinichi Matsuda, Tetsuro Tamaki, Takayuki Asahara, Haruchika Masuda

**Affiliations:** 1Department of Physiology, Tokai University School of Medicine, 143 Shimokasuya, Isehara 259-1193, Kanagawa, Japan; k-sekine629@dokkyomed.ac.jp (K.S.); tamaki@is.icc.u-tokai.ac.jp (T.T.); 2Department of Pediatrics, Dokkyo Medical University, 880 Kitakobayashi, Mibu, Shimotsugagun, Tochigi 321-0293, Japan; 3Department of Regenerative Medicine Science, Tokai University School of Medicine, 143 Shimokasuya, Isehara 259-1193, Kanagawa, Japan; kawaguchi101060@gmail.com (A.T.K.); asa777@is.icc.u-tokai.ac.jp (T.A.); 4Department of Health Management, School of Health Studies, Tokai University, 4-1-1 Kitakaname, Hiratsuka 259-1292, Kanagawa, Japan; m.miyazawa@tsc.u-tokai.ac.jp; 5Department of Health and Sports, Niigata University of Health and Welfare, 1398 Shimami-cho, Kita-Ku, Niigata 950-3198, Japan; haruo-hanawa@nuhw.ac.jp; 6Department of Pediatrics, Tokai University School of Medicine, 143 Shimokasuya, Isehara 259-1193, Kanagawa, Japan; matu0831@is.icc.u-tokai.ac.jp

**Keywords:** anti-inflammation, vasculogenic potential, fulminant myocarditis, cell sheet engineering

## Abstract

**Simple Summary:**

Fulminant myocarditis (FM) is a serious inflammatory lesion of the myocardium accompanied by cardiac dysfunction, transitioning to end-stage heart failure. Due to such a difficult pathology, a therapeutic strategy that exerts a steadfast effect has yet to be developed. Blood mononuclear cells (MNCs) have been previously shown to enhance the quality and quantity of cellular fractions (QQMNCs) with anti-inflammatory and vasculogenic potential using the one culture system. The aim of this study was to investigate whether transplantation therapy with hybrid cell sheets of fibroblasts and QQMNCs improves cardiac function in a rat model with experimental autoimmune myocarditis (EAM) induced by purified porcine cardiac myosin. The transplanted hybrid cell sheet exerts cardioprotective effects against EAM, resulting in limited left ventricular remodeling and partially improved cardiac functions due to revascularization, anti-inflammation, and anti-fibrosis. Thus, tissue engineering using hybrid cell sheets of fibroblasts constructed with QQMNCs is expected to provide an effective therapeutic option for patients with severe FM.

**Abstract:**

Fulminant myocarditis causes impaired cardiac function, leading to poor prognosis and heart failure. Cell sheet engineering is an effective therapeutic option for improving cardiac function. Naïve blood mononuclear cells (MNCs) have been previously shown to enhance the quality and quantity of cellular fractions (QQMNCs) with anti-inflammatory and vasculogenic potential using the one culture system. Herein, we investigated whether autologous cell sheet transplant with QQMNCs improves cardiac function in a rat model with experimental autoimmune myocarditis (EAM). Fibroblast sheets (F-sheet), prepared from EAM rats, were co-cultured with or without QQMNCs (QQ+F sheet) on temperature-responsive dishes. QQ+F sheet induced higher expression of anti-inflammatory and vasculogenic genes (*Vegf-b*, *Hgf*, *Il**-10*, and *Mrc1/Cd206*) than the F sheet. EAM rats were transplanted with either QQ+F sheet or F-sheet, and the left ventricular (LV) hemodynamic analysis was performed using cardiac catheterization. Among the three groups (QQ+F sheet, F-sheet, operation control), the QQ+F sheet transplant group showed alleviation of end-diastolic pressure–volume relationship on a volume load to the same level as that in the healthy group. Histological analysis revealed that QQ+F sheet transplantation promoted revascularization and mitigated fibrosis by limiting LV remodeling. Therefore, autologous QQMNC-modified F-sheets may be a beneficial therapeutic option for EAM.

## 1. Introduction

Fulminant myocarditis (FM) is an inflammatory disease of the myocardium that often deteriorates rapidly, culminating in end-stage heart failure (HF), which is a major cause of mortality in cardiovascular diseases [1,2]. FM is known to be caused by various viral or bacterial infections, toxic substances, autoimmune conditions, or adverse medication effects [3]. Currently, due to a lack of effective therapies for FM, the only option available for treatment of end-stage HF, caused by FM, is heart transplantation. However, there are various problems, including the shortage of heart transplantation donors, prolonged need for immunosuppression, and risk of organ rejection. Therefore, novel treatment strategies are urgently required for end-stage HF. Since the late 1990s, stem cell-based regenerative therapy has been developed as an alternative to heart transplantation [4].

Diverse stem cell sources have been reported since the first experimental attempt for cardiac regeneration was performed with skeletal myoblasts [5], for example, skeletal muscle-derived cells, cardiac stem cells, induced pluripotent stem cells (iPSCs), and endothelial progenitor cells (EPCs) [6].

EPCs have been used as a successful cell source for ischemic diseases to recover injured tissues via therapeutic induction of neovascularization [7,8]. EPCs also secrete paracrine factors that play an essential role in cell-to-cell communication, resolution of inflammation, and subsequent recovery [9].

However, regenerative stem cell therapy requires isolation of stem cells from various sources and ex vivo expansion with optimal culture conditions. In the case of EPCs, there is a fundamental problem with their acquisition from peripheral blood mononuclear cells (PBMNCs), because of their very low frequency in the adult peripheral blood (PB) even in healthy adults (0.5% in PBMNCs). Moreover, the EPC fraction declines numerically and functionally in patients with aging or with cardiovascular risk factors, such as diabetes, hypertension, hyperlipidemia, and obesity [10,11]. Therefore, methodologies to ensure adequate quality and quantity of EPCs are required for robust therapeutic efficacy. We have previously reported a serum-free system of co-culturing human naïve PBMNCs without EPC isolation to selectively expand functional EPCs with anti-inflammatory macrophages (M2Φ) and regulatory T (Treg) cells to selectively expand the functional EPC population without prior isolation of EPCs [12]. The serum-free culture system involves five cytokines: stem cell factor (SCF), thrombopoietin (TPO), FMS-like tyrosine kinase 3 (Flt3) ligand, interleukin (IL)-6, and vascular endothelial growth factor (VEGF). The culture system allows naïve PBMNCs to easily convert into regenerative cell populations and selective enhancement of the quality and quantity of cellular fractions with vasculogenic and anti-inflammatory potential. The therapeutic application of grafting these quality and quantity (QQ)-enhanced culture of blood mononuclear cells (QQMNCs) for lesions in degenerative or inflammatory diseases, such as diabetic wound healing, bone fractures, or myocardial infarction, is expected to increase, based on experimental evidence [13,14,15]. In this context, we have previously reported that transplanted QQMNCs can recover cardiac function in a rat myocardial infarction model due to their enhanced anti-inflammatory activity, as well as potential for revascularization [14]. Our collaborators have demonstrated extended survival of cardiac cell sheets derived from allogeneic mouse iPSCs upon fascial co-transplantation with mouse QQMNCs into the dorsal subcutaneous space [16].

Further, considering the application for cardiac diseases except myocardial ischemia, investigation of the effectiveness of QQMNCs for FM is notably significant for devising strategies to decrease the incidence rate of HF.

On the other hand, as an advantageous cell-based therapy for HF, tissue-engineered cell sheet technology has been developed for effective delivery of regenerative cells [17,18], such as murine adipose cells [19], human iPSCs [20], and EPCs [21] into cardiac tissues damaged by ischemia or myocarditis. We have also demonstrated that fibroblast sheets (F-sheet) co-cultured with EPCs recover cardiac function in a rat myocardial infarction model [22]. Moreover, considering their application for cardiac diseases other than myocardial ischemia, investigating the effectiveness of QQMNCs in FM is thought to be particularly significant for developing strategies to decrease the incidence rate of HF.

Experimental autoimmune myocarditis (EAM) induced in rats by porcine cardiac myosin has been established as an FM model leading to end-stage HF [23]. In the present study, we used this rat model to investigate whether transplantation of F-sheets co-cultured with QQMNCs in the hearts of pathological EAM rats ameliorates myocarditis and preserves heart functions in FM.

## 2. Materials and Methods

### 2.1. Study Protocols

The study protocol (Figure 1 and Appendix A) was approved by the Institutional Ethical Committee for Animal Experiments and is described below.

### 2.2. Construction of Rat EAM Model

Male Lewis rats aged 6–8 weeks, weighing 140–160 g (Charles River Laboratories, Tokyo, Japan), were used in this study. Purified porcine cardiac myosin was dissolved in 0.01 mol/L phosphate buffered saline (PBS) and emulsified with an equal volume of complete Freund’s adjuvant (Difco Laboratories, Detroit, MI, USA). On days 0 and 7, 0.05 mL of emulsion, yielding an immunizing dose of 0.25 mg cardiac myosin per rat, was injected subcutaneously into the footpad [23,24] under sevoflurane inhalation (Maruishi Pharmaceutical Co., Osaka, Japan) (Figure 1B,C).

### 2.3. PBMNC Isolation and QQ Culture

Peripheral blood of EAM rat (8–10 weeks) was collected after anesthesia with 2–3.5% sevoflurane (Maruishi Pharmaceutical Co., Osaka, Japan) from the abdominal aorta using a 10-mL syringe containing heparin (Ajinomoto, Tokyo, Japan). PBMNCs were isolated by density gradient centrifugation using the Lymphocyte Separation Solution (Histopaque^®^-1083, Sigma-Aldrich, St. Louis, MO, USA) as previously reported [12,25]. The composition of the QQ culture medium is presented in Appendix A. Isolated PBMNCs were cultured for 5 days at a density of 2.0 × 10^6^/2-mL QQ culture medium per well of 6-well Primaria plates (BD Falcon, San Jose, CA, USA).

#### 2.3.1. EPC Colony Formation Assay

To determine the vasculogenic potential of PBMNCs or QQMNCs, freshly isolated PBMNCs and post-QQ cultured cells were seeded at a density of 0.25 × 10^5^ cells in 35 mm Primaria dishes (BD Falcon, San Jose, CA, USA) in culture medium containing proangiogenic growth factors and cytokines (Appendix A), as described previously [25]. Seven days after seeding the cells, the number of adherent primitive EPC colony-forming units (pEPC-CFUs) and definitive EPC colony-forming units (dEPC-CFUs) were counted separately in a gridded scoring dish (STEMCELL Technologies Inc., Vancouver, BC, Canada) via phase-contrast light microscopy (Eclipse TE300; Nikon, Tokyo, Japan). In this assay, EPCs generally exhibit the two morphologies in the adherent colonies of ‘pEPC-CFU and dEPC-CFU’, depending on the differentiation cascade. The former indicates ‘primitive EPC-CFU’ and the latter ‘definitive EPC-CFU’ based on the difference in their morphological features between the two colonies. pEPC-CFUs were represented by small round-shaped cells with high proliferative potential, while dEPC-CFUs were represented by large spindle-shaped cells with preferentially vasculogenic potential. In this study, two researchers blinded to the identity of the samples individually identified and counted the pEPC-CFUs formed by the round-shaped cells (diameter, 10 μm to 20 μm), and dEPC-CFUs formed by the spindle-shaped cells (longitudinal length, 50 μm to 200 μm) (Figure 2).

#### 2.3.2. Flow Cytometric Analysis

To investigate the expression profile of QQMNCs, PBMNCs and QQMNCs were analyzed by flow cytometry (FCM) as reported previously [25]. The cells were pipetted in 1.5-mL microtubes (Sumitomo, Tokyo, Japan), and the cell density was adjusted to 5 × 10^5^/mL. Thereafter, 1000 μL of FACS buffer was added to each tube, followed by centrifugation at 1500 rpm for 3 min.

Stain No. 1 and 2: The cells were stained with anti-rat antibodies or isotype control antibodies (with volume and dilutions adhering to manufacturer’s protocol), as previously described [12].

Stain No. 3: M1/M2 macrophages were quantified by FCM using Dako’s IntraStain Kit (Dako, Santa Clara, CA, USA), as described previously [12].

All the fluorophore-labeled antibodies used are listed in Appendix A. Data were collected using BD LSRFortessa (Becton Dickinson, Franklin Lakes, NJ, USA) and analyzed using FlowJo version 7.6.5 software (Tree Star, Inc., San Carlos, CA, USA).

### 2.4. Preparation of F, M+F and QQ+F Sheets

QQMNCs sheets alone could not be acquired and needed co-culture with fibroblasts. However, fibroblasts do not only serve as matrix-producing reparative cells, but exhibit a wide range of functions in inflammatory and immune responses, angiogenesis, and neoplasia [26]. Primary dermal fibroblasts were prepared using an explant culture. Fibroblasts from male EAM rats on day 8 (8–10 weeks) were cultured from abdominal and back area skin samples. Briefly, skin samples were washed several times with PBS (Sigma-Aldrich, St. Louis, MO, USA) containing 1% penicillin/streptomycin and cut into several pieces (5 mm thick and wide). Skin pieces were treated with 0.1% collagenase type IA (Sigma-Aldrich, St. Louis, MO, USA) prepared in DMEM containing 7.5% FBS, with gentle agitation for 2 h at 37 °C. Extracted cells were filtered through 70 µm, 40 µm, and 20 µm nylon strainers to remove muscle fibers and other debris. Fibroblasts were stored in liquid nitrogen using a cell preservative solution (Cell Banker 1; Juji Field, Tokyo, Japan) until use. For the F-sheet, 2 × 10^6^ isolated fibroblasts were seeded in 35 mm temperature-responsive culture dishes (CellSeed, Tokyo, Japan); for the M+F sheet, fibroblasts (2 × 10^6^ cells) and blood mononuclear cells (MNCs) (2 × 10^5^ cells) were mixed and seeded together; and for the QQ+F sheet, fibroblasts (2 × 10^6^ cells) and QQMNCs (2 × 10^5^ cells) were mixed and seeded. After 2 days of culture at 37 °C, the temperature-responsive dishes with confluent cell cultures were transferred to another CO_2_ incubator set at 20 °C for approximately 60 min, a treatment in response to which the cell sheets detached spontaneously. Three-layered cell sheets were then stacked by pipetting according to previously described procedures [27].

#### qRT-PCR

We investigated the features of PBMNCs and QQMNC as well as those of the F sheet, M+F sheet, and QQ+F sheet. Although we did not examine the transplantation of M+F sheet, we examined the differences in cell sheets with and without QQ culture of PBMNCs. Briefly, total RNA was isolated from PBMNCs, QQMNCs, and cell sheets using TRIzol reagent (Invitrogen, Carlsbad, CA, USA). Genomic DNA contaminant was digested with DNase I (Thermo Fisher Scientific, Waltham, MA, USA) at 37 °C for 15 min, and total RNA was purified by phenol extraction and ethanol precipitation. 2 μg of purified total RNA was reverse transcribed into complementary DNA (cDNA) using the High Capacity cDNA Reverse Transcription Kit (Thermo Fisher Scientific, Waltham, MA, USA). The cDNA was diluted 2–1280-fold with Milli-Q water, and real-time qRT-PCR was performed using SYBR Green Master Mix (Thermo Fisher Scientific, Waltham, MA, USA) or TaqMan Gene Expression Assays (Thermo Fisher Scientific, Waltham, MA, USA). Gene expression was normalized to the rat *18S* rRNA gene expression levels. The forward and reverse primer sequences used in this assay are listed in Appendix A.

### 2.5. Transplantation of F and QQ+F Sheets

Fifteen days after immunization in the acute phase [23,24], the rats were randomly assigned to three groups and subjected to thoracotomy (Figure 1A,B) involving the following: (1) sham operation without transplantation (sham group); (2) three-layered F-sheets transplantation onto the anterior cardiac wall (F-sheet group); and (3) three-layered QQMNCs + F-sheets transplantation onto the anterior cardiac wall (QQ+F sheet group). During transplantation, animals were anesthetized with 2–4% sevoflurane, and after the procedure, they were orally intubated with a 14-G intravenous catheter and respirated using a ventilator at 10 mL/kg, 60 times per min (Ugo Basile S.R.L.VA, Italy).

#### 2.5.1. Cardiac Catheterization

Cardiac catheterization was performed using a microtip catheter (Miller Instruments Inc., Houston, TX, USA) 15 days after the sheet transplantation. The pressure–volume loop data were recorded under stable hemodynamic conditions. The rats were anesthetized, and chest was opened to insert the catheter and record repeated data on pressure–volume and obtain blood samples according to a previously described procedure [28].

#### 2.5.2. Histological Analysis

Tissue damage and inflammation are important triggers of fibrosis [29]. Histological analysis was performed to evaluate fibrosis and angiogenesis; 15 days after sheet transplantation, the rats were anesthetized, and the thoracic cavity was opened to harvest heart tissue samples. The samples were fixed overnight in 4% paraformaldehyde (PFA) at 4 °C and washed with a graded sucrose (0–25%)/0.01 M PBS. The samples were then immersed in optimal cutting temperature compound and frozen/stored at −80 °C. Subsequently, 7 µm histological sections were obtained. To evaluate fibrotic areas, heart tissue sections were treated with picrosirius red stain (Muto, Tokyo, Japan). The selected fields (left ventricular: LV, base, papillary muscle, apex level) were evaluated using an automatic fluorescence microscope at a magnification of 2–100× (Olympus cellSens Dimension software 2.1, Japan). We analyzed each specimen slide at the cardiac base, papillary muscle, and apex levels. The percentage of fibrotic area in the three lesions within a given field (%) was determined using ImageJ (https://imagej.nih.gov/ij (accessed on 1 Febuary 2018)) and is shown as the average of three lesions. Immunofluorescen was used to evaluate the vascular density. Blood vessels were detected using monoclonal antibodies against-RECA1 (rat endothelial cell antigen 1) (1:1000; MCA970GA, BIO-RAD, CA, USA), a vascular endothelial cell marker. Staining was visualized using Alexa Fluor-594-conjugated goat anti-mouse antibodies (1:500, room temperature for 1 h; polyclonal probes, Abcam 150116, UK). Cardiac muscles were detected using an anti-cardiac troponin I primary polyclonal antibody (Abcam, ab47003). Reactions were visualized using Alexa Fluor-488 conjugated goat anti-rabbit antibodies (1:500, room temperature for 1 h; polyclonal probes, Abcam 150077). Nuclei were counterstained with DAPI (4,6-diamino-2-phenylindole). The blood vessel counts at the three lesions (LV base, papillary muscle, apex levels) at a magnification of 2–100× were determined using ImageJ. The number of RECA1-positive blood vessels is shown as a total of three levels. Spectral analysis was performed using an Axio Imager M2 fluorescence microscope (ZEN 2.3.69.1000, Carl Zeiss, Jena, Germany).

### 2.6. Statistical Analysis

Statistical analysis was performed using IBM SPSS version 26.0 (SPSS, Inc., Chicago, IL, USA). All values are expressed as the mean ± standard error. Statistical significance was defined as a two-sided *p*-value of <0.05. The Mann–Whitney U and Kruskal–Wallis tests were used to compare the data between the two groups and among three to four groups in each assay. Bonferroni’s multiple comparison test between the two groups after the Kruskal–Wallis test was performed as a post hoc test. Data for ESPVR were compared using the Mann–Whitney *U* test with Bonferroni adjustment, following Kruskal–Wallis analysis of variance (ANOVA). *p*  <  0.01 was considered statistically significant.

## 3. Results

### 3.1. QQMNCs in EAM Rats Retained the EPCs Colony-Forming Potential Similar to That in Healthy Rats

The calculated total QQMNCs derived from 100 mL of peripheral blood decreased from original cell counts (healthy control, cell counts × 10^5^ = 1671 ± 405 to 517 ± 82, *p* < 0.001; EAM, cell counts × 10^5^ = 1572 ± 314 to 815 ± 148, *p* < 0.01), with an average decrease of 0.31-fold in the healthy control rats and 0.28-fold in the EAM rats (Figure 2A). To evaluate the vasculogenic potential of cultured QQMNCs compared to freshly isolated PBMNCs, 0.25 × 10^5^ cells were seeded onto a methylcellulose-coated dish, and at day 7 post-seeding, pEPC-CFUs and dEPC-CFUs were separately counted (Figure 2B,C). The total number of EPC-CFU increased in QQMNCs compared to that in PBMNCs (healthy control, 26.0 ± 5.9 vs. 2.8 ± 1.7, *p* < 0.001; EAM, 35.6 ± 3.7 vs. 3.3 ± 5.4, *p* = 0.002) (Figure 2C). The differentiation ratios of colony-forming EPCs indicated by dEPC-CFUs/total EPC-CFU of the healthy control group were 76.9% and 37.4% for QQMNCs and PBMNCs, respectively, and those of the EAM group were 73.0% and 16.9% for QQMNCs and PBMNCs, respectively (Figure 2D). Moreover, colonies in QQMNCs of the healthy control group were enriched 2.06-fold for dEPC-CFUs compared with those in PBMNCs, and colonies in QQMNCs of the EAM group were enriched 4.32-fold for dEPC-CFUs compared with those in PBMNCs. These data show that QQMNCs increased the number of dEPC-CFUs, which further differentiated the EPC cells for vasculogenesis.

### 3.2. The Restored Cellular Phenotype of QQMNCs Obtained from EAM Rats Was Similar to That from Healthy Rats

FCM analysis revealed that QQ culture had higher frequency of hematopoietic stem cell (CD34, CD133) populations than the PBMNCs in both healthy control and EAM groups (healthy control, 1.38-fold in CD34^+^ cells, *p* = 0.040, 0.55-fold in CD133^+^ cells, *p* = 0.055; EAM, 3.42-fold in CD34^+^ cells, *p* = 0.011, 2.18-fold in CD133^+^ cells, *p* = 0.818). The population of endothelial lineage cells (CD31) was higher than the PBMNCs in the EAM group (healthy control, 1.01-fold, *p* = 0.818; EAM, 5.26-fold, *p* = 0.002). The population of EPCs (VEGFR-2^+^/CD34^+^) had also expanded compared to PBMNCs (healthy control, 5.93-fold in VEGFR-2^+^/CD34^+^ cells, *p* = 0.180, 1.88-fold in VEGFR-2^+^/CD34^+^ cells, *p* = 0.036). The population of CD4^+^/CD3^+^ T cells in the healthy or EAM groups, showed an increasing trend, and increased in QQMNCs compared to PBMNCs (healthy control, 1.10-fold in CD4^+^/CD3^+^ cells, *p* = 0.055; EAM, 1.45-fold in CD4^+^/CD3^+^ cells, *p* = 0.015). In contrast, the population of CD8^+^/CD3^+^ T cells in the both groups, significantly decreased in QQMNCs compared to PBMNCs (healthy control, 0.27-fold in CD8^+^/CD3^+^ cells, *p* = 0.002; EAM, 0.60-fold in CD8^+^/CD3^+^ cells, *p* = 0.002). Additionally, the population of dendritic cells (CD11b^+^/CD11c^+^) did not decrease in healthy rats, but it was significantly decreased in the EAM group (healthy control, 1.36-fold, *p* = 0.485; EAM, 0.12-fold; *p* = 0.002). The population of dendritic cells (CD11b^+^/CD11c^+^) was significantly decreased in the EAM group (healthy control, 1.36-fold, *p* = 0.485; EAM, 0.12-fold; *p* = 0.002). The population of M2 macrophages (CD163^+^, CD206^+^/CD68^+^) had significantly increased in both groups (healthy control, 12.68-fold in CD163^+^ cells, *p* = 0.040, 10.20-fold in CD206^+^/CD68^+^ cells, *p* = 0.002; EAM, 28.34-fold in CD163^+^ cells, *p* = 0.002, 19.29-fold in CD206^+^/CD68^+^ cells, *p* = 0.002). The population of regulatory T cells (CD25/CD3+CD4) had slightly increased in both groups (healthy control, 1.16-fold in CD25^+^/CD3^+^+CD4^+^ cells, *p* = 0.631; EAM, 1.27-fold in CD25^+^/CD3^+^+CD4^+^ cells, *p* = 0.476). These findings indicate that QQMNCs in EAM rats restored the expanded cellular phenotypes with vasculogenic and anti-inflammatory activities, similar to healthy rats (Figure 3 and Appendix A).

### 3.3. The Profiles of Gene Expression of Sheet-Free QQMNCs in EAM Rats

Next, we analyzed the characteristics of gene expression in QQMNCs versus PBMNCs derived from EAM rats. The expression of genes encoding vascular endothelial growth factors (*Vegf-a* [30] and *Vegf-b* [31]) was significantly higher in QQMNCs than in PBMNCs. The increases in QQMNCs versus PBMNCs were 4.12-fold for *Vegf-a* (*p* < 0.01) and 56.08 for *Vegf-b* (*p* < 0.01). The expression levels of cardioprotective factors, including *Vegf-b* [32], hepatocyte growth factor (*Hgf*) [33], and insulin-like growth factor 1(*Igf*-1) [34], were higher in QQMNCs than in PBMNCs. The increases in QQMNCs versus PBMNCs were 3.58-fold for *Hgf* and 2.34-fold for *Igf-1*. The expression levels of anti-inflammatory genes encoding the M2 macrophage biomarker, mannose receptor (*Mrc1/Cd206*) [35] were significantly higher in QQMNCs than in PBMNCs. The increases in QQMNCs versus PBMNCs 4.75-fold for *Mrc1/Cd206* (*p* < 0.05). These findings correspond to the expansion of M2 macrophages in QQMNCs versus PBMNCs. The decreases were 0.11-fold for *Il-10* (*p* < 0.01).

On the other hand, the decrease was 0.21-fold for a master gene of anti-inflammatory Treg cells, forkhead box P3 (*Foxp3*). Moreover, among the genes related to cardiac fibrosis, those encoding matrix metalloproteinases (*Mmp-2* and *Mmp-9*) are prominent [36]; the expression level of *Mmp-2* increased significantly in QQMNCs compared to that in PBMNCs (140.69-fold, *p* < 0.01) whereas expression level of *Mmp-9* was unchanged (1.03-fold increase). We also analyzed the expression levels of genes encoding inflammatory cytokines including *Il**-17*, tumor necrosis factor (*Tnf*), and *Il**-1β.* Expression levels of *Il**-17* and *Tnf* were higher in QQMNCs than in PBMNCs. The increases in QQMNCs versus PBMNCs were 35.45-fold for *Il**-17* (*p* < 0.01) and 4.12-fold for *Tnf* (*p* < 0.01). On the other hand, the expression level of *IL-1β* in QQMNCs was lower than that in PBMNCs (0.20-fold decrease).

Together, the gene expression for vascular regeneration and cardioprotection (*Vegf-a, Vegf-b, Hgf, Igf-1*) was generally enhanced, whereas the inflammatory (*Il-17, Tnf, Il-11b*) and anti-inflammatory genes (*Mrc1/Cd206, Il-10, Foxp3*) were differentially altered in QQMNCs compared to the PBMNCs. In addition, the expression of *Mmp-2*, which is related to cardiac fibrosis, was especially upregulated in QQMNCs. All the data are shown in Figure 4 and Appendix A.

### 3.4. Favorable Conversion of Gene Expression Profiles in QQ+F Sheet against EAM

We also explored the gene expression profiles of QQ+F sheet derived from EAM rats (Figure 5, Appendix A). First, with respect to the genes related to vascular regeneration and cardioprotection, expression level of *Vegf-b* was significantly higher in QQ+F sheet than in F-sheet and M+F sheet. The increases were 1.73-fold (*p* < 0.05) in QQ+F versus F-sheet and 1.84-fold (*p* < 0.05) in QQ+F versus M+F sheet. The expression levels of genes encoding cardioprotective factors *Hgf and Igf*-*1* were higher in QQMNCs than in PBMNCs. *Hgf* was significantly upregulated in the QQ+F sheet than in the F-sheet (2.25-fold, *p* < 0.05). *Igf -1* expression was marginally increased in QQ+F sheet compared to that in F sheet and M+F sheet (*p* = 0.11). The increases were 1.55-fold in QQ+M versus F sheet and 1.68-fold in QQ+F versus M+F sheet, respectively.

Second, the expression levels of genes encoding anti-inflammatory factors (*Mrc1/Cd206*, *Il**-10*) were also upregulated. The increases were 2.85-fold for *Mrc1/Cd206* (*p* < 0.05) in QQ+F versus F-sheet and 1.41-fold in QQ+F versus M+F sheet (*p* = 0.65). *Il**-10* expression was significantly higher in the QQ+F sheet than in F-sheet and M+F sheet, which was different from the profile of sheet-free QQMNCs and PBMNCs. The increases were 2.44-fold for *Il**-10* (*p* < 0.05) in QQ+F versus F-sheet and 2.24-fold in QQ+F versus M+F sheet (*p* = 0.59).

Third, the expression profiles of genes encoding inflammatory cytokine factors (*Il**-17, Tnf*, and *Il**-1β*) when compared between QQ+F and F-sheet, were different from those between sheet-free QQMNCs and PBMNCs. *Il-17* gene expression was not significantly different between the groups (*p* = 0.065). However, the decreases were 0.85-fold in QQ+F versus F-sheet and 0.75-fold in QQ+F versus M+F sheet. The differential gene expression profile of *Il-17* between QQ+F and M+F sheet was inversely changed compared to that between the sheet-free QQMNCs and PBMNCs (Figure 4 and Figure 5, Appendix A). *Il-17* gene expression in sheet-free QQMNCs was significantly enhanced compared to that in PBMNCs, while the gene expression level in QQ+F sheet was comparable to that in M+F sheet. Similarly, the gene expression level of *Tnf* in sheet-free QQMNCs was higher than that in PBMNCs. However, the expression level of *Tnf* was unchanged in QQ+F sheet compared to the M+F sheet, although the levels in both sheets were slightly upregulated compared to that in F-sheet alone. The increases were 2.42-fold (*p* = 0.29) in QQ+F sheet versus F-sheet and 0.69-fold (*p* = 0.16) in QQ+F sheet versus M+F sheet.

Fourth, high expression of genes encoding matrix metalloproteinases (*Mmp*s) *Mmp-2* and *Mmp-9* induce cardiac fibrosis [36]. In particular, the *Mmp-2* gene expression level in the Q+F sheet settled down to the level of M+F sheet, although the levels in both sheets were slightly upregulated compared to F-sheet alone. The increases were 1.59-fold (*p* = 0.15) in QQ+F sheet versus F-sheet and 1.20-fold (*p* = 0.53) in QQ+F sheet versus M+F sheet. In addition, *Mmp-9* gene expression was slightly upregulated in both the QQ+F sheet and M+F sheet versus F-sheet only. In contrast to *Mmp-2*, the gene expression profiles of *Mmp-9* in the Q+F sheet versus M+F sheet did not show drastic profiles like those in QQMNCs versus PBMNCs. The increases were 1.90-fold (*p* = 0.40) in QQ+F sheet versus F-sheet and 1.36-fold (*p* = 0.54) in QQ+F sheet versus M+F sheet.

The findings revealed that the expression of genes related to vascular regeneration (*Vegf-a, Vegf-b*), cardioprotection (*Vegf-a, Vegf-b, Hgf, Igf-1),* and anti-inflammation (*Mrc1/Cd206, Il-10*) were synchronously induced in the cell sheet of QQMNCs co-cultured with fibroblasts (QQ+F sheet). Additionally, the profiles of inflammatory cytokine genes, *Tnf* and especially *Il-17*, which were enhanced in the sheet-free QQMNCs, were subdued when QQMNCs were co-cultured with fibroblasts in cell sheets (Figure 4 and Figure 5, Appendix A).

### 3.5. QQ+F Sheets Transplantation Maintained Healthy Diastolic Cardiac Function

Cardiac catheterization using a conductance catheter revealed the effect of the end-systolic pressure–volume relationship (ESPVR) and end-diastolic pressure–volume relationship (EDPVR) on volume load. EDPVR describes LV compliance. A decrease in LV diastolic function due to a decrease in cardiac compliance is indicated by an upward shift in EDPVR. Conversely, an increase in cardiac compliance is indicated by a downward shift in EDPVR (Figure 6C). ESPVR describes LV contractility.

ESPVR was significantly different between sham and healthy control groups (*p* = 0.009) (Figure 6A,D and Appendix A). On the other hand, the EDPVR on volume load was significantly higher in the sham and fibroblast transplant groups than in the healthy control group (*p* = 0.010 vs. sham, *p* = 0.004 vs. fibroblast group; Figure 6B,E and Appendix A). However, there was no significant difference between the QQ+F sheet transplant and healthy control groups (*p* = 0.206). The findings indicate that QQ+F sheet grafts in EAM hearts did not improve systolic cardiac function, but maintained healthy diastolic cardiac function.

### 3.6. QQ+F Sheets Grafts in EAM Hearts Inhibited the Bnp Gene Expression Indicating HF

BNP is considered a diagnostic and prognostic biomarker for HF [37]. *Bnp* mRNA levels in the heart at 14 days after sheet transplantation were significantly higher in the sham and F-sheet groups than in the healthy control group (*p* = 0.012 for vs. sham, *p* = 0.030 for vs. F-sheet; Figure 7). However, there was no significant difference between the QQ+F sheet transplant and the healthy control group (*p* > 0.05). The findings indicate that QQ+F sheet grafts in EAM hearts did not improve systolic cardiac function, but maintained healthy diastolic cardiac function.

### 3.7. QQ+F Sheets Grafts Limited LV Remodeling in EAM Hearts

Typical histological features of LV remodeling, such as collagen accumulation and capillary density were assessed in the LV using picrosirius red staining (Figure 8A) and immunohistochemistry for anti-RECA1 (Figure 8B), respectively. The average collagen deposition in three heart sections (cardiac base, papillary muscle, and apex) showed a significant decrease in the QQ+F sheet group compared with that in the sham and F-sheet groups (sham, 33.7 ± 3.7%, *p* = 0.015 vs. QQ+F sheet; F-sheet, 29.3 ± 2.5%, *p* = 0.012 vs. QQ+F sheet; QQ+F sheet, 14.8 ± 3.0%; Figure 8C). The average cardiac troponin deposition in the three heart sections (cardiac base, papillary muscle, and apex) showed a significant increase in the QQ+F sheet group compared with that in the sham and F-sheet groups (sham, 24.4 ± 7.2%, *p* = 0.017 vs. QQ+F sheet; F-sheet, 19.2 ± 3.0%, *p* = 0.031 for vs. QQ+F sheet; QQ+F sheet, 30.3 ± 5.3%; Figure 8C). The total blood vessel counts in the three heart sections were significantly higher in the QQMNC group than in the fibroblast group (sham: 11927 ± 1863, *p* = 0.017 for vs. QQ+F sheet, F-sheet: 10238 ± 2357, *p* = 0.417 vs. QQ+F. sheet, QQ+F sheet: 13775 ± 2351; Figure 8C). These findings show that the grafted QQ+F sheets in EAM hearts promoted neovascularization and limited the replacement of myocardia with fibrous tissue.

## 4. Discussion

In the present study, we demonstrated that PBMNCs derived from pathological EAM rats, under vasculogenic and anti-inflammatory culture conditions, may convert to cellular phenotypes with enhanced vasculogenic and anti-inflammatory potential. Moreover, QQMNCs co-cultured with F-sheets may optimize the cardioprotective effects by limiting the expression of genes that deteriorate rat EAM (*Il-17* or *Mmp-2*). As a result, the grafting of QQMNCs co-cultured with F-sheets can contribute to cardioprotective functions in EAM hearts.

### 4.1. Boosted Vascular Regenerative and Anti-Inflammatory Cellular Phenotypes of QQMNCs from EAM Rats

We observed that the anti-inflammatory and vasculogenic potential of QQMNCs was boosted in EAM rats to a level equivalent to that observed in healthy rats (Figure 3 and Figure 4). As previously reported, chronic excessive secretion of inflammatory cytokines in cardiovascular diseases causes impaired EPC bioactivities [10,11]. Of note, EPC-CFA of QQMNCs from EAM rats revealed an enhanced expansion of dEPCs and more differentiated EPCs. dEPCs preferentially exert higher vasculogenic activity than highly proliferative pEPCs and immature EPCs [12,38]. Considering the differential features of the two types of EPCs, the fact that the colony-forming potential of dEPCs even in QQMNCs from EAM rats were similar to those from healthy rats, which suggests favorable vascular regeneration by dEPCs in EAM rats (Figure 2). The population of EPCs (VEGFR-2^+^/CD34^+^) in QQMNCs in EAM rats expanded compared to that in PBMNCs, although the expansion ratio was not significantly higher than in healthy rats. Moreover, the population of anti-inflammatory M2 macrophages (CD206^+^/CD68^+^) in QQMNCs from EAM rats significantly expanded, whereas populations of immunogenic dendritic cells (CD11b/c^+^) and T cells (CD8^+^/CD3^+^), associated with tissue injury, tended to decrease in QQMNCs. The expression of the Treg master gene, i.e., *Foxp3* was downregulated in QQMNCs from EAM rats (Figure 4 and Appendix A); however, there was no change in the Treg phenotype (CD25^+^/CD3^+^/CD4^+^; Appendix A). Previous reports have shown that QQMNCs enhance Treg population or upregulate *Foxp3* expression in healthy rats and humans [12,14]. The different features of Tregs observed in this study might be due to the severe inflammatory environment in the acute phase of EAM. However, *Foxp3* expression in QQMNCs was restored in the cell sheets of QQMNCs co-cultured with fibroblasts, although this was not statistically significant.

Overall, these data indicate that QQ culture conditioning, even in EAM rats, induces features of EPCs and M2 macrophages in PBMNCs with anti-inflammatory and vasculogenic potential similar to that in healthy rats.

### 4.2. Manifested Cardioprotective Gene Expression of QQ+F Sheet in EAM Rats

F-sheet co-cultured with QQMNCs synchronously facilitated the expression of genes-encoding potent cardioprotective factors (*Vegf-b* [32], *Hgf* [33] and *Il-10* [39]), compared to the F-sheet, while co-culture with PBMNCs did not (Figure 5, Appendix A). In QQMNCs from EAM rats, the gene expression of *Il-10* was downregulated compared to PBMNCs. Moreover, the gene expression of inflammatory cytokines (*Il-17* and *Tnf*) in sheet-free QQMNCs was significantly upregulated compared to PBMNCs, whereas in the cell sheet, it declined to levels similar to those in PBMNCs + F sheet. IL-17 is a potent pathophysiological inducer of severe inflammation in EAM [40,41,42]. Considering the crosstalk between *IL-17* and *IL-10* in EAM rats, the inverse relationship of *IL-17* and *IL-10* expression between sheet-free QQMNCs and the cell sheets of QQMNCs co-cultured with fibroblasts may be explained by the inhibitory effect of IL-10 on IL-17 [43,44].

On the other hand, the gene expression of the inflammatory cytokine *IL-1β* in sheet-free QQMNCs tended to be downregulated, and inversely, the cell sheets of QQMNCs co-cultured with fibroblasts enhanced the expression of *IL-1β*. IL-1β in EAM is thought to be a secondary reactive inflammatory cytokine, as IL-1β secretion increases around the lesion according to disease progression [45]. Cardioprotective effects were manifested by the grafted sheets in EAM hearts despite the high expression of *IL-1β* in the F-sheet with QQMNCs. These findings indicate the importance of IL-17, rather than IL-1β, as an essential factor in the pathogenesis of EAM. In addition, the cardioprotective effect of the F-sheet with QQMNCs might be due to the inhibitory effect of IL-10 on IL-17.

Taken together, grafting of the cell sheets of QQMNCs co-cultured with fibroblasts appears to be an advantageous and effective methodology for EAM hearts, when compared with the simple injection of QQMNCs per se.

### 4.3. Favorable Cardioprotective Efficacy of QQ+F Sheets Grafting in EAM Hearts

In the present study, the sham and F-sheets grafting in EAM rats significantly deteriorated the diastolic function (EDPVR), while QQ+F sheets grafting restored conditions to the level in healthy rats (Figure 6). Moreover, the expression of *Bnp,* reflecting HF severity, was upregulated in the EAM hearts of the sham and F-sheet transplant groups, but not in those with QQ+F cell sheet grafts (Figure 7), and histochemical analysis revealed that the grafted QQ+F sheets in EAM hearts accelerated vascularization, reducing myocardial loss and cardiac fibrosis. Together, QQ+F sheets grafting in EAM rats exerted cardioprotective effects by limiting LV remodeling and restoring healthy diastolic function.

### 4.4. Limitation of This Study

In this study, QQ+ F sheets grafting restored the cardiac diastolic function in EAM rats, but not the systolic function. Considering the partial efficacy, the optimal number of QQMNCs-co-cultured fibroblasts required to achieve the best cardioprotective potential for EAM remains to be determined. Alternatively, the QQ+ F sheets were grafted during the period of completely constructed EAM, when its effectiveness might be weakened or incomplete. Considering this, the optimal time for cell sheet grafting also remains to be determined. In this respect, grafting at an earlier stage of EAM could be more beneficial. In a future study, we are planning to determine the optimal time and/or the number of QQMNCs, and advance cell-sheets instead of fibroblasts, e.g., cardiomyocyte-sheets. Further, in vitro phenotypic comparison of M+F and QQ+F sheets derived from the same PBMNCs could be performed, while, due to the time lag of QQ culture period, the simultaneous grafting of those sheets under the same in vivo condition is inevitably impossible to precisely compare the effectiveness of their transplanted sheets. Alternatively, an individual in vivo experiment using frozen PBMNCs and/or QQMNCs for their simultaneous transplantation is required to precisely compare the effects of transplanted fibroblast sheets co-cultured with their cells.

## 5. Conclusions

Transplantation of autologous QQ+F sheets limits cardiac damage in a rat EAM model by exerting cardioprotective effects involving revascularization, anti-inflammation, and anti-fibrosis, resulting in partial restoration of cardiac functions. Collectively, the hybrid cell sheet technology with QQMNCs may pave the way for clinical application as an effective therapeutic option for patients with FM, leading to the suppression or delaying of the onset of HF.

## Figures and Tables

**Figure 1 biology-11-00106-f001:**
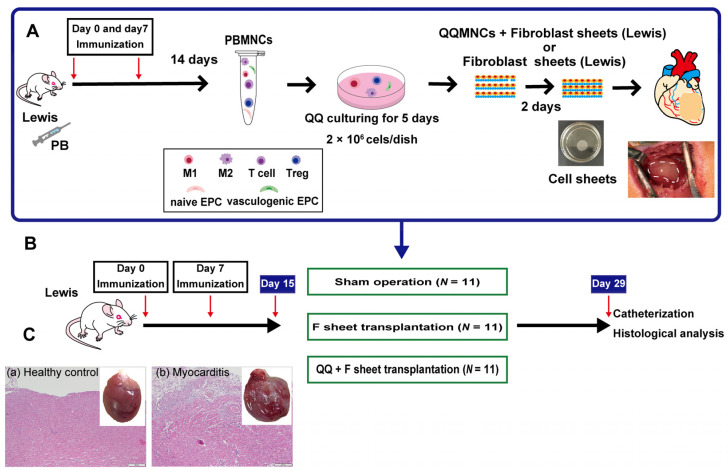
Experimental protocol. (**A**) Schematic flowchart of the in vitro protocol; Differentiation and purification of QQMNCs from rat peripheral blood and three harvested QQMNCs and fibroblast sheets (F-sheets). (**B**) Schematic flowchart of the in vivo protocol; Immunization and transplant of QQMNCs and F-sheets. (**C**) Representative images of the entire heart and heart tissue section stained with H&E at 15 days after immunization with procaine cardiac myosin (a) Healthy control (×4), (b) EAM (×4), Scale bar = 200 μm. PB: peripheral blood, PBMNCs: peripheral blood mononuclear cells, QQMNCs: quality and quantity control culture of mononuclear cells, H&E: Hematoxylin and eosin stain, M1: M1 macrophage, M2: M2 macrophage, Treg: regulatory T cell, EPC: endothelial progenitor cell.

**Figure 2 biology-11-00106-f002:**
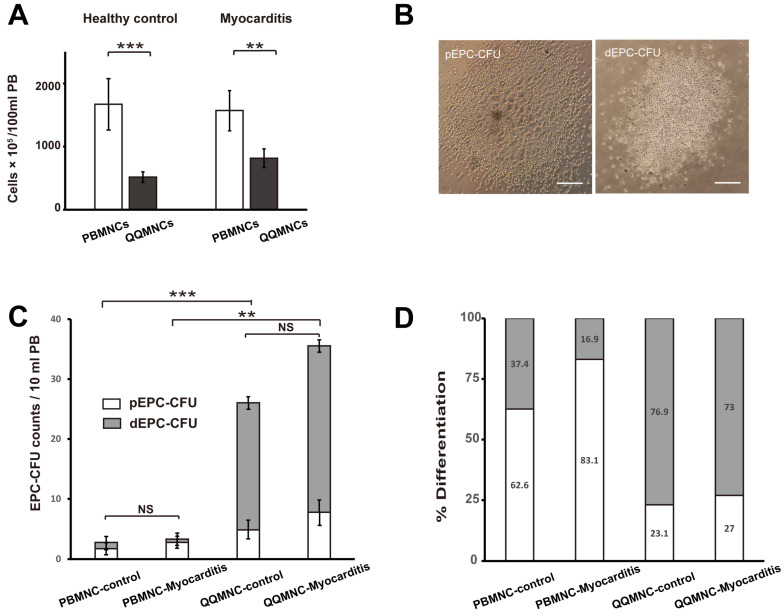
QQ culture in healthy control and rats with myocarditis. (**A**) The graph shows the total cell number of PBMNCs and QQMNCs isolated from 100 mL of peripheral blood. Cell counts showed significant differences between PBMNCs and QQMNCs derived from healthy controls and rats with myocarditis. Cell counts showed no significant difference in QQ culture between healthy controls and rats with myocarditis. Data are shown as the mean ± SE. *N* = 7 to 8. ** *p*  <  0.01, *** *p*  <  0.001, NS: not significant. (**B**) Representative picture of pEPC-CFU and dEPC-CFU in rats with myocarditis. Scale bar = 500 μm. (**C**) The graph shows EPC-CFU counts generated from PBMNCs and QQMNCs per 10 mL of peripheral blood in healthy controls and rats with myocarditis. Comparative analysis showing an increased number of EPC-CFU in QQMNCs of rats with healthy controls and myocarditis. (**D**) The graph shows pEPC-CFU and dEPC-CFU counts as a percentage of total EPC-CFU count per dish. Data are shown as mean ± SE. *N* = 6. ** *p*  <  0.01, *** *p*  <  0.001.

**Figure 3 biology-11-00106-f003:**
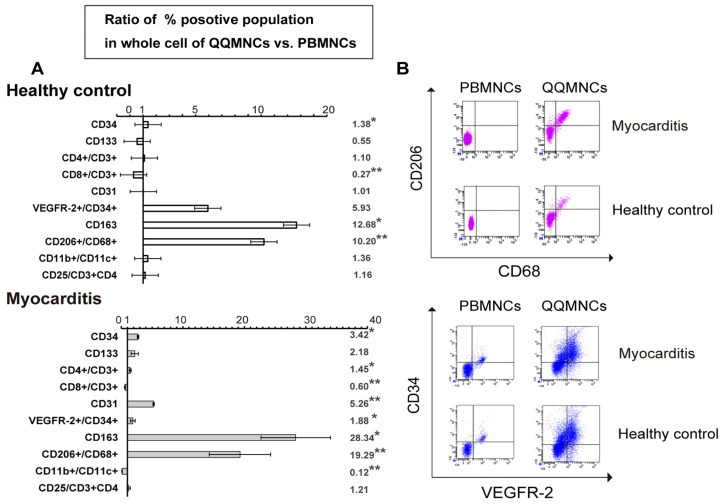
Flow cytometry analysis of QQMNCs derived from rats with EAM. (**A**) The upper bar graph in healthy control shows the ratio of each (%) cell positivity in QQMNCs relative to that in PBMNCs. The lower bar graph in EAM shows the ratio of each (%) cell positivity in QQMNCs relative to that in PBMNCs. The investigated cell surface markers were as follows: hematopoietic stem cell (CD34, CD133), endothelial cell (VEGFR-2+/CD34, CD31), T cell (CD3, CD4, CD8), dendric cell (CD11b/c), monocyte (CD68), M2 macrophage (CD163, CD206/CD86), and regulatory T cell (CD25/CD3+CD4). (**B**) FCM was performed on gated PBMNCs and QQMNCs (gated CD68, CD206, VEGFR-2, and CD34). Data shown as mean ± SE. *N* = 6, * *p*  <  0.05, ** *p*  <  0.01. VEGFR: vascular endothelial growth factor receptor.

**Figure 4 biology-11-00106-f004:**
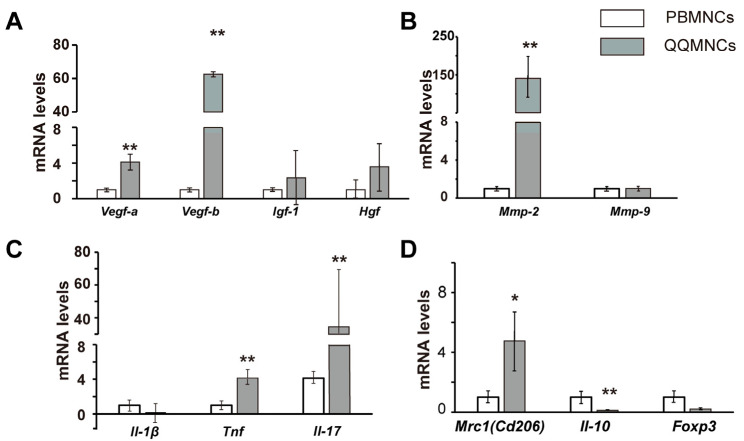
qRT-PCR analysis of PBMNCs and QQMNCs derived from EAM rats. Data are shown as mean ± SE. *N* = 5 or 6, * *p* < 0.05, ** *p* < 0.01 vs. QQMNCs. Enhanced gene expression for vascular regeneration, anti-inflammatory, and cardioprotective effects in a rat model of myocarditis. (**A**) Proangiogenic growth and cardiogenic factors. (**B**) MMPs. (**C**) Inflammatory cytokines. (**D**) Anti-inflammatory cytokines and immune tolerance. Data are shown as mean ± SE. *N* = 5 or 6 per group. VEGF: vascular endothelial growth factor, HGF: hepatocyte growth factor, IL: interleukin, MRC1: mannose receptor (CD206), Igf-1: insulin-like growth factor 1, Mmp: matrix metalloproteinase, Tnf: tumor necrosis factor.

**Figure 5 biology-11-00106-f005:**
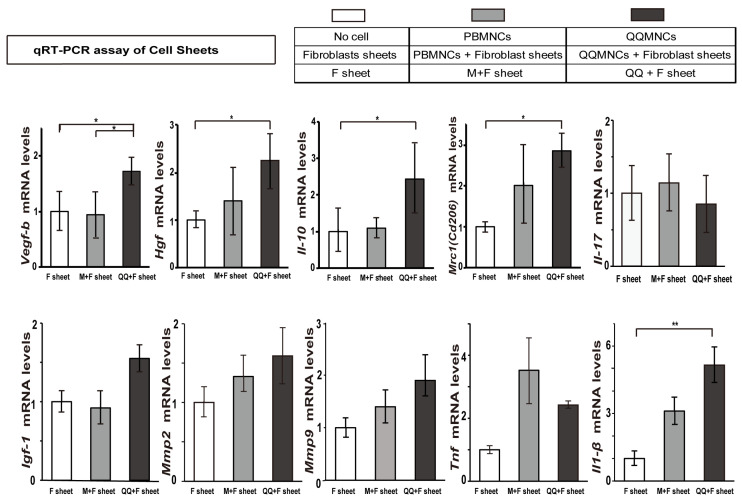
Analysis of gene expression profile in various F sheet with PBMNCs or QQMNCs derived from the EAM rats. QQMNCs were obtained from peripheral blood of the rat EAM model. Expression of genes encoding vascular regeneration, anti-inflammation, and cardioprotection factors (*Vegf-b*, *Hgf*, *Il-10*, *Cd206/Mrc1*) were upregulated in the QQ+F sheet. Inflammatory gene expression (*Il-17*) was downregulated in the QQ+F sheet. Inflammatory gene expression (*Il1-**β*) was upregulated in the QQ+F sheet. Data are shown as mean ± SE. *N* = 5 per group. * *p* < 0.05, ** *p* < 0.01. QQ+F sheet group. qRT-PCR: quantitative real-time polymerase chain reaction, M + F sheet: co-culture sheets of fibroblasts and PBMNCs, QQ+F sheet: co-culture sheets of fibroblasts and QQMNCs.

**Figure 6 biology-11-00106-f006:**
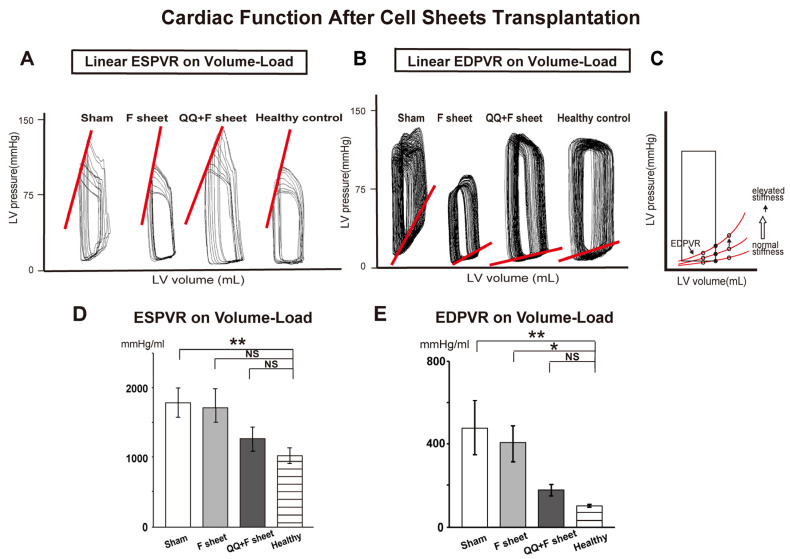
Cardiac function on 15 days after transplantation of QQ+F sheets derived from EAM rats. Cardiac functions were recorded using a conductance catheter and the pressure–volume (P-V) loop was measured as described in Methods. (**A**,**B**) Representative P-V loops on 14 days after cell sheet transplantation. (**A**) The oblique lines indicate end-systolic pressure–volume relationship (ESPVR). (**B**) The oblique lines indicate end-diastolic pressure–volume relationship (EDPVR). (**C**) An schematic diagram showing hypothetical EDPVR. EDPVR describes LV compliance. A decrease in LV diastolic function owing to a decrease in cardiac compliance is indicated by an upward shift in EDPVR. (**D**) The ESPVR on volume load showed significant differences between sham and healthy control groups. *N* = 6 rats per group. (**E**) The EDPVR on volume load were significantly higher in the sham and F-sheet groups than in the healthy control group. There was no significant differences between the QQ+F sheet and healthy control groups. *N* = 6 rats per group. Data are shown as mean ± SE. NS, not significant, * *p* < 0.05, ** *p* < 0.01, vs. healthy control group.

**Figure 7 biology-11-00106-f007:**
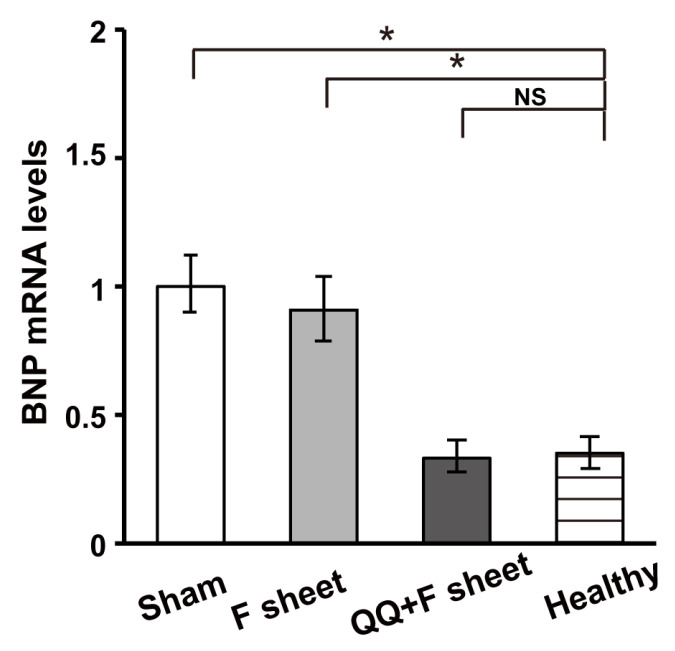
The levels of *Bnp* gene expression in the heart at 14 days after transplantation of QQ+F sheets derived from EAM rats. *Bnp* mRNA levels, as measured by qRT-PCR, were significantly higher in the sham and F-sheet groups than in the healthy control group. There was no significant differences between the QQ+F sheet and healthy control groups. *N* = 5 rats per group. Data are shown as mean ± SE. * *p*  <  0.05, vs. healthy control group. NS, not significant. *Bnp*, brain natriuretic peptide.

**Figure 8 biology-11-00106-f008:**
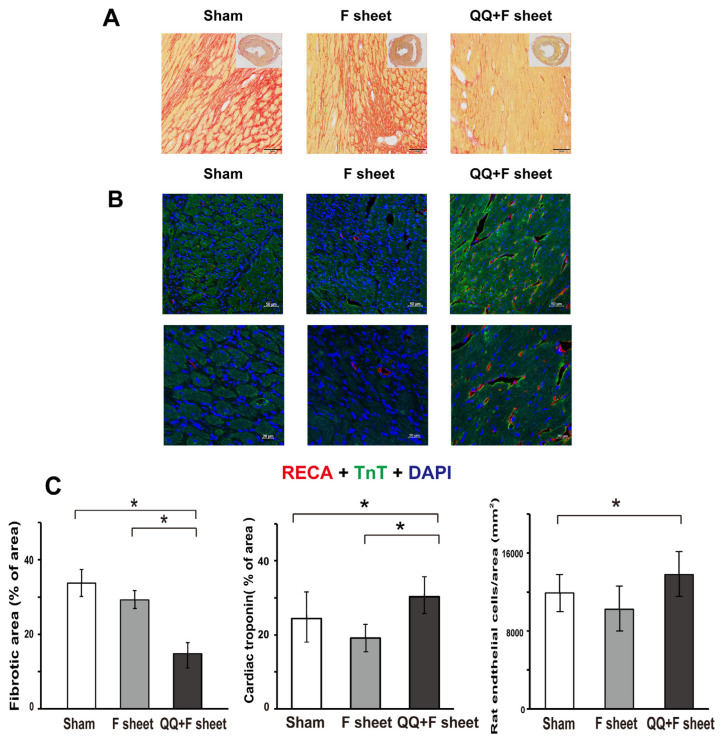
The effect of QQ+F sheets transplantation on left ventricular (LV) remodeling in the hearts of EAM rats. Heart tissues were harvested 14 days after cell sheet transplantation and markers for LV remodeling were detected using histological analysis, as described in Methods. The QQ+F sheet transplant group showed significantly reduced heart fibrosis. (**A**) Representative heart sections stained with picrosirius red (×2 and ×100) at 14 days after sheet transplantation. To evaluate fibrotic area, the heart tissue sections were treated with picrosirius red stain. The percentage of fibrotic area was determined using ImageJ software, and the threshold color plugin set to red fibrotic tissue was selected and converted to a binary image. Scale bar = 100 μm. (**B**) Representative heart sections were stained with immunofluorescence RECA-1(×20; Scale bar = 50 μm, ×40; Scale bar = 20 μm) at 14 days after sheet transplantation. RECA (red): anti-endothelial cell antibody (RECA-1), TnT (green): anti-cardiac troponin T antibody, DAPI (blue). (**C**) The average collagen deposition of three heart sections (cardiac base, papillary muscle, and apex) showed a significant decrease in the QQ+F sheet group compared with that in the sham and F-sheet groups (*p* < 0.05). Number of RECA1-positive blood vessels per mm^2^. The total blood vessel counts of three heart sections showed an increase in the QQ+F sheet group compared with that in the F-sheet group (*p* < 0.05). The average cardiac troponin deposition of three heart sections (cardiac base, papillary muscle, and apex) showed a significant increase in the QQ+F sheet group compared with that in the sham and F-sheet groups (*p* < 0.05). Data are shown as mean ± SE. *N* = 6 rats per group. * *p* <  0.05; vs. QQ+F sheet group.

## Data Availability

All data generated in the present study are available in the manuscript and Appendix A.

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
