# Peer review of "Transplantation of Fibroblast Sheets with Blood Mononuclear Cell Culture Exerts Cardioprotective Effects by Enhancing Anti-Inflammation and Vasculogenic Potential in Rat Experimental Autoimmune Myocarditis Model"

_biology, 2022, doi:10.3390/biology11010106_

Round 1

Reviewer 1 Report

.

Author Response

Thank you for your valuable suggestions on our manuscript.

Reviewer 2 Report

The under-review article entitled “Transplantation of Fibroblast Sheets with Blood Mononuclear Cell Culture Exerts Cardioprotective Effects by Enhancing Anti-Inflammation and Vasculogenic Potential in Rat Experimental Autoimmune Myocarditis Model” by Sekine et al has been reviewed. This research reports how the cell sheet engineering can effectively improve the cardiac function. The results and work is quite interesting and but it need the further revision to consider for the publication.

As the COVID 19 vaccine cause the myocarditis problems, is it the present stem cell can help to address the present issue? authors have any opinion about it?

The text in the line 30-31 must be deleted in the abstract.

The introduction text is well described and highlighted the importance of the work in better and understanding manner.

M& M has been presented well that provided the sufficient information that required for the reproduction of the work, but minor grammatical and spell check is required.

Fig.2B, the scale bar is not visible, and the cell grouping is not clear, which need the high-resolution figure to compare the results with presented text.

Fig.2A need to show the statistical difference between the QQNBCs and PBMNCs.

Fig.3.,4D&E,Fig.6 C showed the + SE but not able to see the -SE? author should revise the figure to enable both values in the bars. In raw the last figure is not statistically different with treatments.

Author Response

(The authors gave the same response as above.)

Reviewer 3 Report

The authors investigated the significance of autologous cell sheet transplant with quality and quantity-enhanced culture of blood mononuclear cells (QQMNCs). Authors’ results are remarkable to identify the property of QQMNCs. It has a potential to improve cardiac function in a rat model promoting revascularization and mitigating fibrosis by limiting LV remodeling. However, the reviewer has some concerns about the description of results. Please consider and reply for comments and questions.

Major

  1. Authors explained a lot in results, but some figures were in supplementary figures. Please transfer Figure S1 and S2 to main manuscript. In addition, please add the histogram about Igf-1, Tnf, Il-1B, and Mmp2/9 in Figure 3. They can support authors’ explanations to understand.
  2. Did authors investigate M+F sheet transplantation study about Figure 4, 5, and 6? That group is important to identify the significance of QQMNCs.

Minor

  1. Please show the material and method about Figure S1 in supplementary materials.
  2. The description in line 270 was incorrect. CD4+ T cells increased in experimental autoimmune myocarditis (EAM) significantly, and CD8+ T cells decreased significantly.
  3. The description in line 273 was incorrect. Dendric cells had significantly decreased in EAM.
  4. The description in line 329 was incorrect. QQ+F sheet was not upregulated significantly compared to M+F sheet.
  5. Please explain about Figure 4C in results.
  6. How many tissue slides did authors analyze with each section (cardiac base, papillary muscle, and apex)? Please add in materials and methods.

Author Response

(The authors gave the same response as above.)

Round 2

Reviewer 1 Report

Well done ! I really appreciated your answers. Congratulation on your work.

Reviewer 3 Report

Authors replied sincerely and correctly. I would like to accept in present form.